# Navigating the risks: Stakeholder views on risk-based cervical cancer screening

**Maali-Liina Remmel** [1]*, **Kadri Suija**[1,2], **Anna Markina**[3], **Anna Tisler**[1], **Anda Ķīvīte-Urtāne**[4], **Mindaugas Stankūnas**[5], **Mari Nygård**[6], **Gunvor Aasbø**[6,7], **Laura Maļina**[4], **Anneli Uusküla**[1]

**1** Institute of Family Medicine and Public Health, Faculty of Medicine, University of Tartu, Tartu, Estonia, **2** Institute of Public Health and Clinical Nutrition, Faculty of Health Sciences, University of Eastern Finland, Kuopio, Finland, **3** Faculty of Social Sciences, School of Law, University of Tartu, Tallinn, Estonia, **4** Institute of Public Health, Riga Stradiņš University, Riga, Latvia, **5** Department of Health Management, Lithuanian University of Health Sciences, Kaunas, Lithuania, **6** Department of Research, Cancer Registry of Norway, Oslo, Norway, **7** Department of Interdisciplinary Health Science, University of Oslo, Oslo, Norway,

\* maali-liina.remmel@ut.ee (MLR)

## Abstract

### Background

The development of risk-based cancer screening programs requires a paradigm shift in existing practices and healthcare policies. Therefore, it is crucial to not only assess the effectiveness of new technologies and risk prediction models but also to analyze the acceptability of such programs among healthcare stakeholders. This study aims to assess the acceptability of risk-based cervical cancer screening (RB CCS) in Estonia from the perspectives of relevant stakeholders.

### Methods and materials

This qualitative study employed semi-structured interviews with healthcare policy and service level stakeholders in Estonia. The Theoretical Framework of Acceptability guided the interview design, and the findings were charted using framework analysis based on the Consolidated Framework for Implementation Research.

### Results

17 interviews were conducted with stakeholders, including healthcare professionals, cancer registry representatives, technology specialists, policymakers, and health insurance providers. While stakeholders generally supported the concept and potential benefits of RB CCS, recognizing its capacity to improve screening outcomes and resource allocation, they raised significant concerns about feasibility, complexity, and ethical challenges. Doubts were expressed about the readiness of the healthcare system and population, particularly the current health information system's capacity to support risk-based approaches. The need for evidence-based and internationally validated screening models, comprehensive public communication, provider training, and collaborative discussions involving all relevant parties, including the public, was emphasized.

**Data availability statement:** The data underlying this study contain indirect identifiers that may risk the identification of individuals, even if deanonymized. To protect participant confidentiality, the data will not be publicly shared. However, deanonymized data may be made available upon reasonable request. The data is in Estonian language. Requests for access to the data should be directed to the Tartu University Research Ethics Committee at eetikakomitee@ut.ee.

**Funding:** This work was made possible by the funding received from the EEA (European Economic Area) and Norway Grants under grant EMP416 and grant number PRG2218 from the Estonian Research Council. The grant EMP416 was obtained by A.U., A.K-U., M.S., and M.N. https://eeagrants.org/. The grant nr PRG2218 was obtained by A.U. (https://www.etis.ee/portal/projects/display/c5e5394a-4e4d-44f7-ac9e-cae5dea1139b). Funders did not play a role in study design, data collection and analysis, decision to publish, or preparation of the manuscript.

**Competing interests:** The authors have declared that no competing interests exist.

## Conclusion

The favorable attitude towards RB CCS among stakeholders provides a strong foundation for advancing its development. However, a comprehensive strategy emphasizing the generation of robust evidence, strengthening healthcare infrastructure, prioritizing patient empowerment, and cultivating a collaborative environment built on trust is crucial.

## Introduction

Cancer screening paradigms are evolving, with risk-based approaches- categorizing participants into groups based on risk factors beyond age and sex and providing tailored screening strategies- gaining notable attention for their potential to optimize efficiency and balance costs, harms, and benefits. [1–6] However, shifting towards personalized screening involves considerable challenges related to essential changes in policy, organization, and service delivery. [7] In addition, the integration of high-quality health data is paramount in the development and refinement of risk prediction models, and its safe and ethical utilization must be ensured.

To effectively implement these changes in real-world scenarios, early involvement of patients, healthcare providers, public health organizations, researchers, health insurance providers, patient advocacy groups, technology companies, policymakers, and data donors such as electronic health record managers, is crucial. [7,8]

Yet, the perspectives and concerns of the key stakeholders remain underexplored. [9] Existing research on risk-based cancer screening acceptability is predominantly limited on breast cancer, with an emphasis on women and screening providers (healthcare professionals), while there is limited investigation of other cancer types and stakeholder groups who play a pivotal role in building a comprehensive strategy for risk-based cancer screening. [10–16]

To our knowledge, no studies have explored risk-based cervical cancer screening (RB CCS) acceptability among comprehensive group of stakeholders.

### Cervical cancer screening

Since their inception in Europe during the late 1950s and early 1960s, organized cervical cancer screening (CCS) programs have been widely implemented and are considered among the most successful cancer screening initiatives. [17–19] However, low coverage rates in many regions, persistent disparities affecting minorities and vulnerable populations, the widespread occurrence of opportunistic testing, and the objective to balance harms and benefits have collectively fueled a growing interest in developing risk-based screening programs for cervical cancer. [1,20–25]

In light of the evolving landscape of CCS, this study seeks to fill the research gap by exploring the perspectives of a comprehensive group of healthcare service and policy level stakeholders in Estonia on the acceptability of designing and implementing risk-based cervical cancer screening (RB CCS) programs.

## Methods and materials

### Design

This qualitative study was conducted within an interpretivist research paradigm [26] and involved semi-structured interviews with stakeholders to gather data and framework method for analysis. [27]

We employed the definition of acceptability of healthcare interventions developed by Sekhon et al as our foundation to engage in discourse with healthcare policy and provision level stakeholders [28] and chose the Consolidated Framework for Implementation Research (CFIR, 2022 version) to guide the analysis and reporting of results allowing a comprehensive analysis of individual and contextual factors. [29]

According to Estonian legislation, ethics approval was not required since the study did not involve patients, focused on professional knowledge within public roles and no personal sensitive data or medical data was colleceted. The study posed minimal risk to participants.

[30–32] This study adhered to the highest ethical standards throughout the research process. All data collected were pseudonymized during analysis to ensure participant confidentiality and privacy. Additionally, the principles of informed consent were strictly followed, with oral consent obtained from all participants prior to their involvement. Participants were fully informed about the study's purpose, procedures, risks and benefits, confidentiality safeguards and their rights, including the option to withdraw at any time without consequences. These measures ensured that the research was conducted in a transparent, ethical, and legally compliant manner in accordance with Estonian legislation and institutional policies.

The decision to obtain only verbal consent from healthcare providers for interviews about their professional work was based on several considerations: the research poses minimal risk, focuses on professional experiences without delving into sensitive personal issues, strict confidentiality was ensured and voluntary participation assured.[33] Written consent was deemed unnecessary to reduce administrative burden and based on our experience, healthcare providers and policymakers often prefer a less formal process for minimal-risk studies. International guidelines allow verbal consent in low-risk research, especially where written consent may be unduly burdensome. [33] The consent process was documented through recorded audio statements confirming their participation.

This study followed Consolidated Criteria for Reporting Qualitative Research (COREQ). [34]

## Setting

In Estonia, a country with high cervical cancer mortality, the CCS program has undergone multiple reforms since 2021, including the shift from Pap smears to primary HPV testing, the broadening of eligibility to all women between the ages of 30 and 65, irrespective of their health insurance coverage, and the addition of a home-testing option to the screening program. [35]

Estonian health policy is led by the Ministry of Social Affairs (MoSA), with financing from the Estonian Health Insurance Fund (EHIF). [36] Healthcare services are provided by various entities under MoSA, local municipalities, or private ownership. The MoSA also oversees agencies such as the National Institute for Health Development (NIHD), responsible for the Estonian Cancer Screening registry. CCS is a joint effort of EHIF and NIHD, guided by a National Cancer Screening Group with representatives from relevant organizations. Testing is carried out by midwives, gynaecologists, or GPs, with opportunistic testing also prevalent. This organizational setting guided our selection of stakeholders for the study.

## Participants

This study engaged healthcare service and policy level stakeholders (SPSs) from various levels of the Estonian healthcare system.

A combination of purposive and snowball sampling was employed: initial participants were selected, who then recommended other relevant professionals. We aimed for maximum variation by including stakeholders from diverse organizational and service levels. Of the

28 professionals invited via email, 17 agreed to participate. Recruitment continued until it became clear that further data collection would offer diminishing returns in increasing conceptual depth. Saturation was therefore not treated as a distinct event but rather as a gradual process, evaluated through the researchers' ongoing and cumulative judgment. [37]

## Data collection

Data was collected through in-depth semi-structured individual interviews. The interview guide was collaboratively developed (AK-U, GA, KS, MS, AT, MN, LM, AU, MLR) using the Theoretical Framework of Acceptability (TFA) as a foundation.[28,38] TFA defines acceptability as a multifaceted construct that reflects how appropriate an intervention is perceived to be by those delivering or receiving it, based on cognitive and emotional responses. It includes seven constructs: affective attitude, burden, perceived effectiveness, ethicality, intervention coherence, opportunity costs, and self-efficacy. According to Sekhon, acceptability judgments can be formed by both patients and professionals before experiencing an intervention. [28]

Before the interviews, participants received an informative presentation on RB CCS and were introduced to two hypothetical scenarios involving women with different risk levels. Details of the interview guide are available in the S1 File Interview guide.

Three female researchers (MLR, KS, AT) conducted the interviews—two medical doctors and a public health researcher, all experienced in qualitative interviews. Interviews were conducted via web-based platforms (Teams or Zoom) for participant convenience, digitally recorded, transcribed verbatim, and de-identified. No repeat interviews were conducted. Participants had the opportunity to review and comment on transcripts; one participant did so and confirmed that the transcript accurately reflected her views.

The interviews took place between March 2023 and February 2024.

## Data analysis

We employed framework analysis with combined inductive and deductive approach. [27] The analysis process commenced with open coding of eight transcripts by three analysts (MLR, KS, AM). Two researchers independently coded each transcript, and several discussions were held between the analysts for the purpose of reflexivity and the creation of initial coding structure which was subsequently collaboratively charted into CFIR (by MLR, KS, AM). A "best fit" approach was employed- the constructs most relevant for this study setting were selected and constructs with a significant amount of overlap were merged, while others were adapted to better align with the data they encompassed. This approach enabled us to develop a comprehensive framework that could effectively accommodate the collected data while providing specificity to the review context. [9,29] The "best fit" framework derived from this process was used to deductively code the remaining 9 transcripts by one of the researchers (MLR). However, flexibility was maintained to allow inclusion of new findings and therefore the framework adapted as needed via team discussions, which was also the medium to compile the codes into subthemes under each CFIR construct. We used NVivo software version 12 (QSR International) to analyse the data.

## Consolidated framework for implementation research (CFIR)

CFIR framework was selected for analysing and systematizing the acceptability of risk-based CCS among stakeholders. CFIR is among the most highly cited frameworks in implementation science and its overarching aim is to predict or explain barriers and facilitators to implementation effectiveness. CFIR can be used in formative evaluations in the development phase

of health interventions and has been demonstrated to be a comprehensive and suitable framework for analysing and systematizing the acceptability of risk-based CCS among stakeholders, as demonstrated in a previous systematic review by Taylor et al. [9] CFIR consists of five high level domains: Innovation domain, Outer and Inner setting domains, Individuals domain and Implementation process domain. [29]

Our study utilized two complementary frameworks—the Theoretical Framework of Acceptability (TFA) and the Consolidated Framework for Implementation Research (CFIR)—to address both individual perceptions and broader implementation factors. The TFA guided the design of our interview guide, focusing on constructs such as affective attitude and burden, enabling an in-depth understanding of the acceptability of risk-based cancer screening.

The integration of CFIR was also data-driven, as participants frequently referenced broader contextual factors during interviews, including healthcare infrastructure, policy landscapes, and the implementation process. This dual-framework approach allowed us to balance detailed insights into individual acceptability with a comprehensive understanding of the wider implementation context.

## Reflexivity

We strived to ensure high reflexivity (i.e., self-examination of how researchers' position, background, pre-existing beliefs and shared experiences with the study participants can affect the interviewing and analysis). [39] Our team of researchers consisted of women aged 33–58, three of the researchers are medical doctors (AU, KS, MLR), two epidemiologists (AU, AT) and one social scientist (AM), all of whom have previous experience in conducting qualitative studies. Interviewees did not have a previously established relationship with the interviewers. The regular discussions of the interpretations with our team of researchers supported reflexivity and helped to maintain consistent interpretation.

## Results

We conducted 17 interviews with a duration of 44 to 85 minutes, mean 56 min (SD = 8.6). Participants were aged 26–67, mean age 51 (SD = 11.3) years. Participant characteristics are presented in Table 1.

The best-fit CFIR framework with the subthemes of the constructs is presented on Fig 1.

**Table 1. Characteristics of the participants.**

| Characteristics | | N (%) |
|---|---|---|
| **Gender** | Female | 15 (88) |
| | Male | 2 (12) |
| **Stakeholder group** | G1. Healthcare service providers (gynaecologists and midviwes) | 9 (41) |
| | G2. Cancer screening registry (public health organization) and technology and diagnostics specialists (Centre of Health and Welfare Information Systems (CeHWIS)) [a] | 3 (18) |
| | G3. Policymakers (Ministry of Social Affairs/ Cancer Screening Group of MoSA professionals) | 4 (24) |
| | G4. Health insurance providers (Estonian Health Insurance Fund (EHIF) professionals) | 3 (18) |
| **Years of experience in their field related to provision or organization of cancer screening** [b] | < 5 years | 5 (29) |
| | 5–10 years | 4 (24) |
| | 11–20 years | 3 (18) |
| | > 20 years | 5 (29) |

[a]Joined into one group to prevent identification due to small number of participants.

[b]Or in a role that could potentially be involved in the development or execution of risk-based CCS in the future.

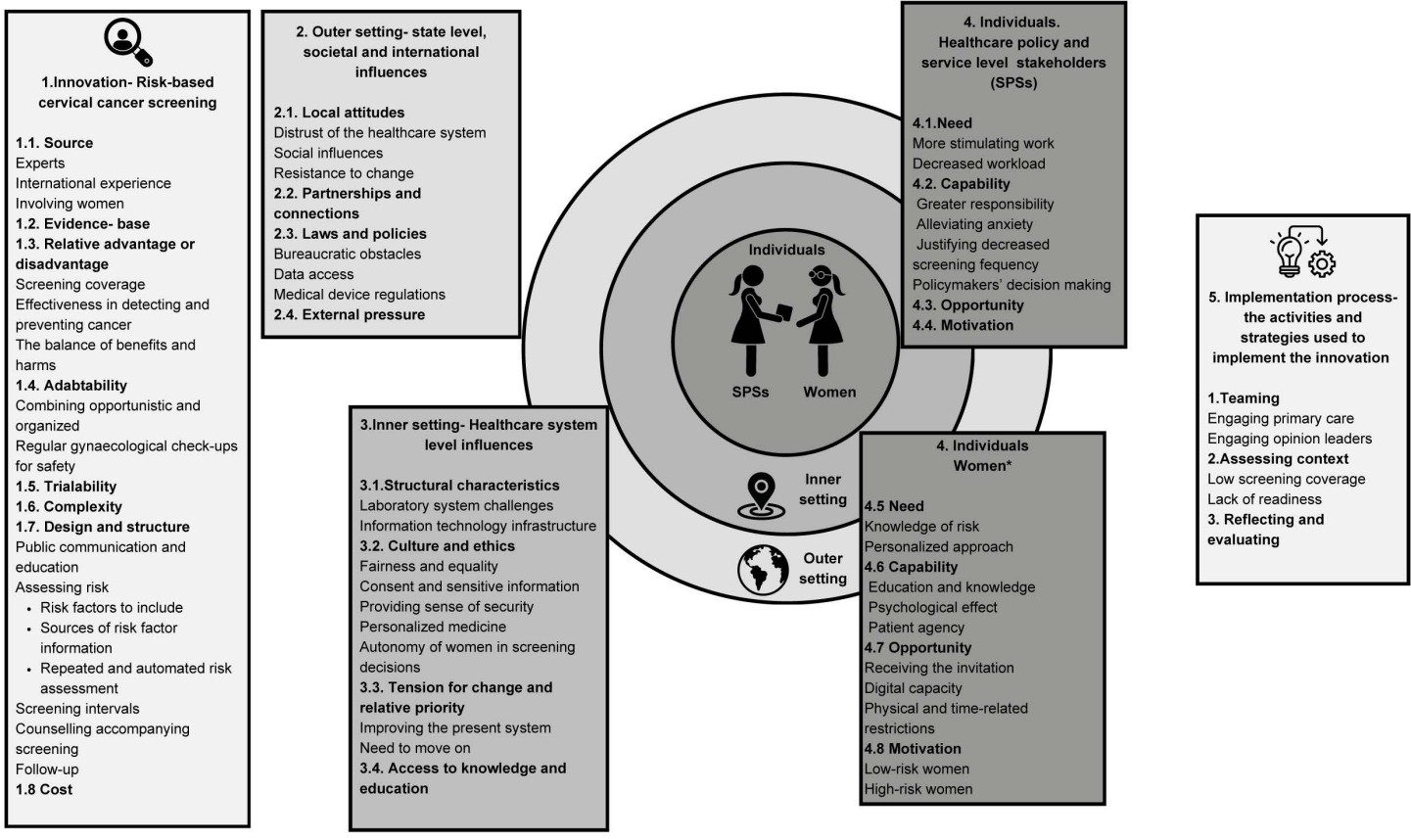

**Fig 1. Healthcare policy and service level stakeholders' perspectives on risk-based cervical cancer screening.**

The best fit CFIR model with subthemes is presented. Subthemes are presented where there was more than one subtheme under the same construct. The figure is adapted from Taylor et al, 2023 [9] and created using Canva.

In the following, the key findings of each of the five domains with their contained constructs are described.

## 1. Innovation- risk-based cervical cancer screening

There was a broad agreement on the need for expert collaboration in designing risk-based programs, learning from international experiences. While the extent of target group involvement in design was debated, strong evidence was deemed crucial for determining risk factors, screening intervals, and cost-effectiveness. Evidence should demonstrate the superiority of personalized screening over the current approach. (Illustrative quotes presented in Table 2: 1.1 and 1.2)

Concerns existed about whether personalized screening would increase coverage, especially among less health- conscious individuals. While RB CCS was perceived effective for early detection and prevention, late diagnoses in low-risk women remain a concern. Many SPSs doubted its feasibility in the current context. While several stakeholders viewed risk-based screening as providing a favorable balance between benefits and harms, others considered the current guidelines to be optimal. (Table 2: 1.3)

**Table 2. Innovation. Subthemes and quotes.**

| Construct | Subthemes | Quotes |
|---|---|---|
| **1.1 Source** | Experts | „There needs to be a working group who establishes and carefully considers it. It's not an individual decision for anyone." (G2) |
| | International experience | "There is no point in inventing a bicycle by ourselves." (G1) |
| | Involving women | "They (patients) often don't see another perspective and want to propagate their worldview. This is a very complex issue; I cannot say what's right." (G1) |
| **1.2 Evidence base** | | "You need a mature idea, which is evidence-based… this is the whole foundation." (G4) |
| **1.3 Relative advantage or disadvantage** | Screening coverage | "For some, this may increase the seriousness of the issue as women feel more effort has been invested and their health data analysed, potentially improving participation." (G3)<br>"If we're talking about a person who simply doesn't care about their health, doesn't even come once in 10 years, then their personal risks also wouldn't mean anything." (G1) |
| | Effectiveness in detecting and preventing cancer | "It would definitely be effective for early detection." (G3)<br>"The risk may be low, but cancer can still go undetected for years." (G1)<br>"If it runs well and the tests are highly specific and sensitive, it reduces mortality and allows for earlier detection. But it depends on planning and execution. One size fits all may prevent more cancer cases in our context." (G4) |
| | The balance of benefits and harms | "It saves healthcare resources, and it's easier for people when they know they don't have to go for check-ups as often. For those with high risk, it helps treat them timely without over-screening low-risk individuals." (G1)<br>"Treatment guidelines have become lenient enough compared to the past, where we often ended up with overtreatment." (G1) |
| **1.4 Adaptability** | Combining opportunistic and organized | "It should be communicated that if she still worries, she can make a doctor's appointment for testing, without charge." (G3) |
| | Regular gynaecological check-ups for safety | "There should definitely be regular visits to the gynaecologist alongside, in that case I believe that ten years between screening tests could be suitable." (G2) |
| **1.5 Trialability** | | "It must be piloted and analysed before nationwide rollout. There should be trials and readiness to act upon the results to change things, play through the alternatives. We should act conservatively, based on evidence." (G4) |
| **1.6 Complexity** | | "Tailor-made solutions improve the system but also increase vulnerability. I question if the game is worth the candles when the current system is robust and functional." (G4)<br>"This [low understanding] can lead to even abstaining from going altogether if it's so confusing." (G2) |
| **1.7 Design and structure** | Public communication and education | "The public advertisement should be as broad as for regular screening and it should be also promoted through general practitioners and gynaecologists who can best explain its importance." (G2)<br>"Preliminary information is definitely important, like why it's done and what I have to gain or lose from it." (G3) |
| | Assessing risk Risk factors to include | "I don't see how someone could be categorized as low risk without vaccination." (G1)<br>"If I am certain that she is not carrying high-risk HPV, only then could such a message of low risk be given to her." (G1)<br>"A genetic test seems more evidence-based than just adding HPV, smoking, and pregnancies… Genes seem more reliable" (G2) |
| | Assessing risk Sources of risk factor information | "If we can extract more from the data being collected to assess risks more precisely, that's definitely something that should be done" (G1).<br>" If trying to draw evidence-based conclusions based on e-health data, errors become quite significant. Combining information from women and databases is necessary" (G3).<br>"That's not possible (using self-reported risk factors). We can't trust that." (G1) |
| | Assessing risk Repeated and automated risk' assessment | "Today she may be in a low-risk group, but maybe in two or three years she won't be anymore. The risk should probably be regularly assessed for the same individual." (G3) |
| | Screening intervals | "We could offer preventive treatment earlier, detect the disease, and provide treatment (if testing high-risk women frequently)." (G1)<br>"When you have seen a patient whose cervical cancer has progressed to the third stage in seven years from the last screening, then of course you start to think now I know that it is possible. This period of ten years seems dubious to me, I would want to take it more often." (G1)<br>"At first, it seems like a big risk (10 year screening interval). If in the Netherlands that they haven't had an increase in interval cancers, then perhaps it's worth considering." (G3) |
| | Counselling accompanying screening | "It needs to be very carefully thought out how to explain it to women so as not to cause fear if the risk is high, and in the case of low risk not to cause carelessness." (G3)<br>"This is not information that could be given via email or SMS. The person will have a lot of questions, I think even telephone counselling is problematic. Such a message can only be delivered directly when meeting with the patient." (G1) |
| | Follow-up | "Testing all women isn't enough; what's crucial is the follow-up process, especially for high-risk cases. Making follow-up as easy as possible is key to the success of risk-based screening." (G4) |

*(Continued)*

**Table 2.** (Continued)

| Construct | Subthemes | Quotes |
|---|---|---|
| **1.8 Cost** | | "Targeting funds more effectively makes sense for state healthcare spending." (G4)<br>"It might require more resources than our current system. It's important to calculate the economic benefits of a personalized system, considering we save on tests some people don't need. But I think this benefit is questionable." (G1) |

A hybrid approach, allowing opportunistic testing, was preferred by many. Some advocated for a shift towards a more flexible system accommodating patient needs. Regular gynecologist visits are seen as an important safety net alongside screening. Piloting risk-based screening before national implementation is recommended. (Table 2: 1.4 and 1.5)

The complexity of the risk-based approach necessitates clear communication and education. Challenges include deciding screening intervals and risk levels, creating guidelines, designing digital solutions, counseling, and obtaining consent. (Table 2: 1.6)

Careful consideration is needed for public communication, risk assessment, screening intervals, counseling, and follow-up. Comprehensive communication strategies are essential, combining public messaging with individualized doctor-patient discussions.

Incorporating vaccination and HPV status, genetics, and family history into risk assessment was emphasized, with debate on including lifestyle factors. Health databases are valuable but potentially incomplete, suggesting a need for supplemental self-reported data and public risk calculators. However, concerns existed about the reliability of self-reported data.

The dynamic nature of risk factors necessitates repeated and ideally automated risk assessments. Many supported shorter intervals for high-risk women but were concerned about extending intervals for low-risk women due to potential missed diagnoses. Some were open to a 10-year interval with sufficient evidence, especially for vaccinated women.

Skilled counselling is essential to address anxiety, ensure understanding, and promote adherence. Face-to-face counseling is preferred for high-risk cases, but alternative options are needed for accessibility. Reliable and accessible follow-up procedures are crucial, particularly for high-risk individuals. (Table 2: 1.7)

While the potential for cost-effectiveness exists, uncertainties remain about resource allocation and potential cost increases with personalized screening. (Table 2:1.8)

## 2. Outer setting- state level, societal and international influences

Low trust in the healthcare system, reliance on personal experiences, and resistance to change in screening practices were seen as challenges for adopting RB CCS in Estonia. (Illustrative quotes presented in Table 3: 2.1)

However, existing collaborations among various organizational tiers could facilitate the development of RB CCS. (Table 3: 2.2)

Legal and regulatory barriers, particularly regarding data access and the classification of risk calculators, also present challenges. (Table 3: 2.3)

Several SPS expressed those external pressures, such as international endorsements and recommendations, play a significant role in the potential acceptance and implementation of RB CCS in Estonia. (Table 3: 2.4).

## 3. Inner setting- Healthcare system level influences

Laboratory system challenges regarding reliability and standardization necessitate quality assurance measures. Developing technological solutions for RB CCS is demanding, with issues arising from both institutional and national databases. The fragmented and incomplete nature

**Table 3. Outer and inner setting. Subthemes and quotes.**

**Outer setting**

| Construct | Subthemes | Quotes |
|---|---|---|
| **2.1 Local attitudes** | Low trust towards healthcare | "There is generally a relatively low trust towards healthcare system among our population as it tends to be the case in countries with colonial or totalitarian histories, often healthcare system is seen as an extension of state structures, and distrust stems from this." (G1) |
| | Social influence | "They come to screen when someone in their circle of friends has been diagnosed." (G1) |
| | Resistance to change | "It has absolutely not been an easy transition (from PAP to HPV testing). Especially older women, who have been going all their lives and ask, why aren't you taking my sample now." (G1) |
| **2.2 Partner-ships & Connections** | | "There is a strong team at Estonian Health Fund, strong input from the university is definitely expected. Early involvement from different institutions like the Health Development Institute, technical solutions parties, laboratories is integral." (G4) |
| **2.3 Laws and policies** | Bureaucratic obstacles | "Bureaucratic steps take more time than creating the technical solution...certain groups of people have created healthcare solutions, but these simply don't reach nationwide use because bureaucracy kills off these services." (G2) |
| | Data access | "We need to think about which legal body can do the risk calculation, they have to have the right to view women's data." (G2) |
| | Medical device regulations | "There are very strict regulations in the field of medical devices...studies must prove that the calculator works in a clinically significant way and meets clinical and diagnostic standards." (G2) |
| **2.4 External pressure** | | "The stepwise transition to risk-based screening is a general recommendation from the European Commission, therefore I am in favour of this." (G4)<br>"I see that in the future we have personalized screening, but for that, there should be recommendations coming either from the World Health Organization or from a higher level" (G2) |

**Inner setting**

| Construct | Subthemes | Quotes |
|---|---|---|
| **3.1 Structural characteristics** | Laboratory system challenges | " If we are assessing a woman's risk based on her previous HPV test results, we can't be sure if the labs all the where HPV tests are done match the national screening program's standards." (G1) |
| | Information Techno-logy Infrastructure | "It's not simple, IT development and changing work processes to account for individual differences in intervals." (G4)<br>"I know our hospital's data goes into the national system, but private clinics don't always do this. Even if a woman knows her test details from a private gynaecologist, I might not find the results because they're in the clinic's private archives. It's very tedious and complicated." (G1)<br>"If an individual has no data in the population registry, she does not get the invite." (G4)<br>"This is a future-oriented thing, so data acquisition is getting better and better, I think for women who are a couple of generations younger than me this will be possible. By that time, the health information system could probably have all features already so advanced, serving as aids for decision-making." (G2) |
| **3.2. Culture and ethics** | Fairness and equality | "It should not be that those who haven´t added anything (data on risk factors) would be left out of the game." (G4)<br>"It needs to be considered how to reduce unequal treatment in terms of varying quality of counselling." (G1) |
| | Consent and sensitive information | "I think it should be asked of every woman whether she wants it or not." (G3)<br>"Women may have worries about someone accessing their data without consent, strong opposition from a data privacy standpoint can arise." (G2) |
| | Providing sense of security | "For me, the benefit of screening is peace of mind... they have that sense of security that when it's positive they will receive help and in the case of a negative result it means everything is fine." (G4)<br>"There might be a feeling that you are somehow letting the patient down, limiting their ability to check (if they request opportunistic testing)." (G1) |
| | Personalized medicine | "It (risk-based screening) fits very well with our concept of moving towards more personalized healthcare service and that is very positive." (G2)<br>"If Estonia truly wants to be a pioneer in personalized medicine, then we should quickly establish a completely new health information system." (G1) |
| | Autonomy of women in screening decisions | "While direct coercion might seem unacceptable, it may be necessary. Linking screening participation to benefits an individual receives could be an effective motivator."(G1) |
| **3.3 Tension for change and relative priority** | Improving the present system | "The first step should be optimizing the existing system, as current coverage is still low. We should focus more on educating people instead of developing risk-based screening." (G1) |
| | Need to move on | "We have a lot of cervical cancer... launching and developing such a program (RB CCS) is crucial. It's in the interest of our women, the entire society, and the children born in the future. A matter of national importance. I've heard about fixing the current program for years. It's not practical because we can never perfect the system entirely. We need to inform people more about risk factors to increase participation beyond 70–80%. We must develop a new screening methodology and educate people simultaneously." (G3) |
| **3.4 Access to knowledge and education** | | "Training should precede this, including general knowledge about how it is developed and additional psychological or counselling training... We definitely need much deeper and greater preparation for a risk-based approach." (G1) |

of the e-health system, unstructured data, inaccessible information, and lack of integration between databases pose hindrances. Despite these challenges, some stakeholders expressed optimism about future possibilities, particularly those involved in e-health development. (Illustrative quotes presented in Table 3: 3.1)

While most stakeholders perceived differential screening intervals as fair, concerns existed about ensuring voluntary participation, preventing exacerbation of inequalities related to data access and digital literacy, and ensuring equitable access to quality counseling services.

The concern about digital literacy specifically related to fears that not all women would be equally able to navigate digital tools, such as using an online risk calculator, which could lead to disparities in participation. The majority supported obtaining consent for participation in risk-based screening, though some suggested it might not be practical and omitting consent could be acceptable if it serves the individual's best interests. However, lack of consent could raise privacy concerns. Maintaining a sense of security through screening is important for patient satisfaction and trust. There is a strong desire for personalized healthcare, including risk-based screening, despite concerns about its current feasibility. A modernized health information system is needed to support personalized approaches. While women's autonomy in choosing to participate in RB CCS was emphasized, some advocated for measures to increase participation rates, raising ethical considerations. (Table 3: 3.2)

There was tension between prioritizing the development of risk-based screening to address high cervical cancer incidence and low screening rates, and the need to improve the existing program through enhanced communication, follow-up, and potentially expanded eligibility criteria. (Table 3:3.3)

Comprehensive training and guidelines for healthcare providers, especially in counseling and risk assessment, are crucial for successful implementation. (Table 3:3.4)

## 4. Individuals

**Service and policy level stakeholders.** Some healthcare providers stated that a shift to RB CCS could make their work more stimulating and decrease workload. (Illustrative quotes presented in Table 4: 4.1) However, many found the implementation challenging at the patient level, particularly in explaining less frequent testing for low-risk individuals, especially in light of the recent transition to HPV testing. However, some healthcare providers expressed confidence in implementing and counselling under a risk-based approach.

Interviewees questioned their ability to effectively manage anxiety related to risk information in counseling, with healthcare providers directly involved with women expressing greater doubts than those not directly involved.

Policymakers were confident in transitioning to RB CCS with sufficient evidence and well-planned strategies. (Table 4: 4.2)

While time constraints for counseling are a concern, potential solutions like providing additional reading materials or employing extra support staff were discussed. (Table 4: 4.3)

The success of RB CCS implementation depends heavily on healthcare providers' motivation to implement changes, and some interviewees expressed potential non-compliance if not fully convinced about its efficacy and safety. (Table 4: 4.4)

**Women as the screening target group.** It's important to emphasize that these insights represent stakeholder perspectives, not direct women's voices.

Some stakeholders were confident that women would appreciate personalized screening and knowing their cervical cancer risk. (Illustrative quotes presented in Table 4: 4.5)

**Table 4. Individuals. Subthemes and quotes.**

**Service and policy level stakeholders**

| Construct | Subthemes | Quotes |
|---|---|---|
| **4.1 Need** | More stimulating work | "I think this makes counselling more substantive, because you're not just the one performing the analysis and delivering the answer, it makes the work more interesting." (G1) |
| | Less workload | "If she continues in the screening program, it means that the workload of gynaecologists would decrease." (G1) |
| **4.2 Capability** | Greater responsibility | "There will be a greater responsibility falling on healthcare workers as one message for everyone doesn't work anymore." (G4)<br>"It doesn't trouble me that I have to start counselling people individually." (G1) |
| | Justifying decreased screening frequency | "It's quite difficult to persuade them otherwise (not needing opportunistic testing). I'm telling you, it's impossible." (G1) |
| | Alleviating anxiety | "Counselling is extremely difficult, so that she wouldn't panic." (G1)<br>"I think the doctor can definitely instill and provide reassurance."(G2) |
| | Policymakers' decision making | "When there is good groundwork and analyses have been done, then I imagine that making this decision isn't really anything particularly extraordinary."(G3) |
| **4.3 Opportunity** | | "There's not enough time during the appointment; it definitely can't be covered in twenty minutes. I envision that if the patient reads through the material at home, comes to the appointment with a few specific questions, which I can address, then counselling can proceed in that manner." (G1) |
| **4.4 Motivation** | | "When a person goes to the gynaecologist, the doctor may just go ahead and do all the tests to calm the patient, and that's maybe even a bigger challenge to change this." (G3)<br>"If it's my duty to inform about risk-based screening, I'll do it. However, I'll also express my personal disagreement. If the person is already in consultation and deemed low-risk, I'd suggest we can still do the testing." (G1) |

**Women as the screening target group**

| Construct | Subthemes | Quotes |
|---|---|---|
| **4.5 Need** | Knowledge of risk | "Everyone is interested in these kinds of risks or well, most people." (G1) |
| | Personalized approach | "I particularly like that from the patient's perspective it's more personalized, it seems to me that our current screenings are too impersonal." (G4) |
| **4.6 Capability** | Education and knowledge | "It is quite a difficult topic for women to understand- what it means when a virus is oncogenic, but not all viruses are, and what it means when it persists in my cervical cells, and how long it persists, and what happens next and how eventually cancer develops from this HPV infection, to understand that, one must make an effort." (G1)<br>"As one of my former colleagues said, I go for "screening" every year. I think this reflects how much people actually know." (G4)<br>"From my experience some women feel that when not having a sexual life, or after a certain age, they no longer need to go to a gynaecologist." (G1)<br>"Some people simply ignore it (screening) because they feel physically well." (G1) |
| | Psychological effect | "If the risk is high, it may cause significant health anxiety. When I tell a 54-year-old woman that her risk is high, she almost wants to come in for conization or order a coffin. They easily adopt such extreme anxiety."(G1)<br>"Some of them (if receiving a positive screening test result) go to the women's clinic emergency room, saying, "Please tell me I don't have cancer." So there definitely would be some psychological challenges." (G1) |
| | Patient agency | "We have ª generation growing up that relinquishes responsibility for their health to someone else, medical system or healthcare professionals." (G1)<br>"If I ask women about their last screening, then I see that it immediately evokes guilt, the fear of judgment is quite significant. We have taken away their right to take responsibility and make decisions about their personal health. " (G1)<br>"I think it (RB CCS) definitely encourages women to take more responsibility for their health behaviour." (G2) |
| **4.7 Opportunity** | Receiving the invitation | "Some lack data or emails in the population register, causing invitations to not reach them physically. This impacts both risk-based and current screening." (G1) |
| | Digital capacity | "The situation arises where not everyone adds risk factors, not due to unwillingness, but perhaps due to lack of awareness or digital skills. "(G3) |
| | Physical and time-related restrictions | "Home testing suits women with physical disabilities preventing clinic visits, those avoiding examinations due to negative experiences or psychological barriers, caregivers occupied with sick family members, or individuals with long workdays unable to visit clinics." (G1) |
| **4.8 Motivation** | Low-risk women | "Perhaps some people believe, 'Well, I don't have to go at all as there's no risk, right? (when receiving low-risk result)." (G1) |
| | High-risk women | "If it is more frequent (for high-risk), women may participate less, especially if the last test was fine and they may think 'it's always the same story every time, nothing changes'." (G4) |

Educating women is crucial, as knowledge gaps exist regarding cervical cancer and its prevention. Understanding cervical cancer and its pathogenesis can be challenging. Low understanding of the screening system and limited awareness of preventive measures, especially when asymptomatic or not sexually active, were noted. (Table 4: 4.6)

Informing women about high-risk levels can cause anxiety and concerns were raised about their ability to handle this information. Healthcare professionals who had witnessed patient distress from positive HPV tests were most concerned. Improving women's sense of agency in managing their health is crucial. Personalized screening itself could encourage greater initiative. A non-judgmental approach from service providers was also seen as essential. (Table 4: 4.6)

Potential barriers to participation in RB CCS include not receiving invitations, limited digital skills, and physical or time constraints. Home-testing could alleviate some of these barriers. (Table 4: 4.7)

Concerns were expressed about high-risk women losing motivation for frequent testing and low-risk women being demotivated by a sense of assurance. (Table 4: 4.8)

## 5. Implementation process- the activities and strategies used to implement the innovation

Beyond expert involvement in designing RB CCS, primary care engagement and leveraging opinion leaders are crucial for successful implementation. (Illustrative quotes presented in Table 5: 5.1)

The importance of considering local context was stressed, given low current screening participation and concerns about applying research findings from other countries. Many deemed implementation in the current Estonian situation infeasible, with midwives and gynecologists being more skeptical than policy-level stakeholders. (Table 5: 5.2)

Rigorous monitoring of screening and continuous improvement is essential, with key performance indicators including screening coverage, cancer detection, stage at detection, follow-up adherence, test result reliability, sample-taking quality, and long-term CC mortality, incidence, and patient life expectancy. Some also mentioned assessing patient satisfaction. (Table 5: 5.3)

Table 5. Implementation process. Subthemes and quotes.

| Construct | Subthemes | Quotes |
|---|---|---|
| **5.1 Teaming** | Engaging primary care | "We should engage primary care providers, our lung cancer screening pilot program demonstrated great success in engaging patients through general practitioners. Additionally, having coordinators based in family health centres to convey information to people could be beneficial." (G3) |
| | Engaging opinion leaders | "Getting some politicians on your side is of help in the journey to becoming a national service as this process is just so bureaucratic." (G2) |
| **5.2 Assessing context** | Low screening coverage | "I'm actually a fan of a personal approach, but I fear that despite the excellent experiences and research results of other countries, the interval would be too long because our women are quite underassessed. " (G1) |
| | Lack of readiness | "At present, making that decision is definitely not possible. If someone in Estonia were to start doing it now, I would oppose it vehemently, because we lack the prerequisites for it, we absolutely cannot do it now. I certainly advocate for risk-based screening, there's no question about it, it's a very reasonable approach, but unfortunately, we don't have any readiness for it." (G1) |
| **5.3 Reflecting and evaluating** | Importance of monitoring | "When transitioning to risk-based screening, ensuring quality and monitoring is essential, along with adjusting risk calculation calculations accordingly if necessary." (G2) |
| | Women's feedback | "I'm really sad that often only the effectiveness is looked at, but other quality indicators, such as how satisfied a woman was with her treatment or how she was informed about the screening result, that so little attention is paid to them in Estonia." (G1) |

## Discussion

While the interviewed stakeholders held a positive view towards the concept and future implementation of RB CCS with the recognition of the potential for improved screening outcomes and resource optimization, significant concerns about feasibility, complexity, ethical challenges and doubts about the readiness of the current healthcare system and target group women were expressed. Notable apprehensions were voiced about the capability of the current health information system to support RB CCS. It was highlighted how implementing risk-based screening would require robust evidence, particularly concerning the significance of risk factors in risk assessment and the safety of reducing screening frequency for low-risk groups. The need for comprehensive public communication and education, adaptable screening models, extensive training for healthcare providers with emphasis on communication and counselling skills, and importance of international endorsement was highlighted.

To the best of our knowledge, this study represents the first exploration of RB CCS acceptability among comprehensive group of key healthcare organizers and service providers. Several findings of this study align with prior research on risk-based screening acceptability among healthcare professionals for other cancer types. There is a higher inclination to approve increased screening frequency for individuals at high risk, compared to hesitancy to decrease screening frequency for those at low risk. The importance of strong evidence and clear communication and education for the public, as well as comprehensive preparation of healthcare professionals has also been emphasized previously. [9–12,16,40,41] Our study further revealed a noteworthy perception among stakeholders that information about cervical cancer, with its intricate infectious etiopathogenesis, poses a unique challenge for comprehension among the screening target group compared to other cancer types. Similarly to current study, service infrastructure capability has also been underscored as a potential challenge in implementing risk-based cancer screening. [9,41]

While prior research emphasizes patient involvement in the development of risk-based cancer screening programs [9], our study reveals a hesitancy among stakeholders to fully embrace this paradigm. This reluctance, particularly evident in concerns regarding the reliability of self-reported data and patient autonomy, may be rooted in a lingering paternalistic healthcare culture inherited from Estonia's Soviet past. A possible parallel example can be observed in the NIHD working group, which led reforms in CCS in 2021. Despite including policymakers and various stakeholders, the group notably excluded members of the public. [35] Systemic barriers such as paternalistic healthcare cultures and a lack of acknowledgment of patient expertise can create a 'glass ceiling,' impeding individuals from fully exercising their agency. [42] However, as some SPSs explicitly highlighted the importance of engaging women, a shift appears to be in progress. Fostering a culture that empowers women to confidently navigate risk-based screening with its increased decision junctures is crucial. A well-implemented personalized approach could further enhance patient agency in CCS. [43,44]

Several ethical challenges were highlighted and while adjusting screening frequency based on individual characteristics was not seen as inherently unjust, many SPSs suggested a flexible approach combining national and opportunistic screening to provide women a sense of security if demanded. This highlights the complex ethical landscape of risk-based cancer screening, where defining the balance between beneficence and non-maleficence is difficult. [45] The public's prioritization of perceived benefits over potential harms underscores the need for further discussions to establish a clear and acceptable benefit-to-harm ratio for screening programs [45–49] While recurrent risk assessment and comprehensive counselling may offer reassurance (as suggested by some SPS), broader public dialogue is essential to ensure that risk-based approaches align with societal values and expectations.

In addition, it was emphasised that preventing a risk-based approach from exacerbating inequalities and creating injustice or discrimination requires careful consideration. The key ethical considerations associated with RB CCS include the potential for unfairness in differential screening, the importance of respecting individual autonomy, and the need to accommodate individuals who may not participate in risk assessments. [45]Screening programs have been criticized for exaggerating the advantages, not giving invitees objective information, and subtly pushing them to participate.[50] Although public health programs do not mandate informed consent, its implementation in the context of risk-based cancer screening is discussed.[7,45,50] However, this poses significant challenges, as a stratified approach is inherently more complex and raises concerns about ensuring the screening target group fully comprehends the information to provide informed consent. Additionally, risk stratification introduces further questions, such as whether the risk assessment process should be required for screening eligibility, and how to manage individuals with missing data or digital literacy -such as difficulty using online tools to input risk information- or choose not to provide risk information. [45]

While SPSs suggested that the personalized nature of RB CCS could enhance the credibility of screening invitations, they also emphasized that the success of RB CCS largely depends on the messages delivered by trusted healthcare professionals. Prior studies also indicate that professionals doubtful of the effectiveness of risk-based screening are less inclined to discuss it with eligible women.[51–53]. Notably, policy-level SPSs not directly involved with the screening target group women were generally more optimistic about the feasibility of RB CCS compared to midwives and gynaecologists. This may reflect a tension described by Esquivel-Sada et al. between clinical ethics, which centres on individual needs, and public health ethics, which emphasizes on collective needs. [13] Thus, building trust and confidence among healthcare providers is crucial for the successful implementation of RB CCS. This requires robust evidence, reliable infrastructure, and comprehensive training.

It has also been argued that risk calculators and other personalized risk assessment tools might be less effective in encouraging risk-reduction behaviours than personal narratives and social influences, such as experiences shared by peers and celebrities [52,54]—a concern echoed by SPSs in this study. Therefore, strategies that address psychological factors influencing risk perception—such as emotions, intuition, social comparisons, and identities—might be considered in the design of communication strategies for risk-based screening programs. [54]

While major improvements are clearly needed for adequate infrastructure for risk-based screening, it is possible, that the scale of obstacles might be overestimated by stakeholders to some extent, as perceived barriers do not always correspond to the actual potential for innovation. This could be illustrated by the scepticism of many SPSs toward digital solutions and health databases for use in RB CCS. However, experts directly involved with information technology argued these advancements are feasible soon with some functions already in operational. In addition, Estonia's substantial investment in digital technology, with most government and healthcare administrative functions online, could provide a strong foundation for further developments needed for RB CCS. Similarly, the distrust towards the laboratory results could partly by rooted in the acknowledged low quality of cytology in Estonia in the past [55], which may not necessarily transfer to the automated HPV-tests. These considerations underscore the importance of consulting all relevant parties and leveraging their expertise in developing RB CCS. A degree of conservatism and preference for improving the current system may stem from the recent reforms in cervical cancer screening, resulting in fatigue and resistance to further changes. However, these reforms may also have established a foundation for future policy advancements through improved networks and the formation of the cancer screening group within MoSA. [35]

Cervical cancer screening is unique, being the first organized cancer screening program with no other cancer screening initiative matching its level of success. This long history has entrenched certain paradigms deeply in the minds of both women and healthcare providers, making change difficult. Furthermore, designing risk-based screening programs for cervical cancer poses unique challenges compared to other cancers, such as breast cancer, which has well-defined genetic risk factors, or newer screening initiatives like lung cancer screening, where there is perhaps less resistance to change. Nonetheless, the advent of novel HPV testing offers a promising opportunity for a paradigm shift in cervical cancer screening and embracing risk-based approaches in light of these developments could enhance the effectiveness and efficiency of cervical cancer prevention efforts.

## Strengths and limitations

By including various organizational and service-level stakeholders the study captured a broad spectrum of viewpoints. Given Estonia's relatively compact administrative structure, our study benefited from the ability to engage a diverse range of stakeholders beyond healthcare service providers, offering a more holistic perspective on RB CCS implementation.

It's worth noting that the predominance of women among gynaecologists, midwives, and individuals involved in cancer screening organization and other stakeholder positions in Estonia is reflected in our study sample, which was largely composed of women.

The study's strength lies in its methodological rigor, employing the theoretical framework of acceptability to guide data collection and the CFIR model to analyse the data to ensure a comprehensive understanding of stakeholder perspectives. While not the primary focus of our study, this approach also allowed us to identify potential barriers and facilitators related to RB CCS program development and implementation.

This study had limitations. The study focused on prospective acceptance of risk-based screening, which may differ from actual acceptance and behaviours once the program is implemented. [28] Future research needs to evaluate the real-world impact and acceptability of RB CCS after its implementation. Nonetheless, evaluating anticipated acceptability is crucial, as it can pinpoint aspects of the intervention that could be adjusted to enhance acceptability and create viable RB CCS programs. While the heterogeneous sample enabled us to capture a broad range of viewpoints and we are confident that saturation was largely achieved [37], the small number of participants in some subgroups means that the possibility of additional themes emerging with more respondents cannot be entirely ruled out.

In addition, investigating the views of the screening target group is crucial, but was out of the scope of this study.

Due to its qualitative nature, the findings may not apply to other SPSs or settings. However, the transferability of our findings is supported by using the CFIR as a well-established framework and providing a thorough description of the study's context which allows for evaluating the potential application of the results in other settings. [56]

## Conclusion

While stakeholders held a favorable perspective toward the future adoption of RB CCS, our analysis revealed several concerns regarding perceived risks and implementation challenges in the current context. These apprehensions ranged from issues with digital infrastructure readiness to doubts about the sufficiency of evidence and patient preparedness. The central concern appears to be whether the perceived benefits outweigh the significant investments and uncertainties associated with the transition. Therefore, advancing toward risk-based screening requires a comprehensive strategy that not only focuses on robust evidence generation and

infrastructural enhancements but also prioritizes patient empowerment, fosters healthcare provider engagement, and cultivates a collaborative environment based on trust.

## Supporting information

**S1 File. Interview guide.**
(DOCX)

## Author contributions

**Conceptualization:** Maali-Liina Remmel, Kadri Suija, Anna Tisler, Anda Kivite-Urtane, Mindaugas Stankūnas, Mari Nygård, Gunvor Aasbø, Laura Maļina, Anneli Uusküla.

**Data curation:** Maali-Liina Remmel.

**Formal analysis:** Maali-Liina Remmel, Kadri Suija, Anna Markina.

**Funding acquisition:** Anda Kivite-Urtane, Mindaugas Stankūnas, Mari Nygård, Anneli Uusküla.

**Investigation:** Maali-Liina Remmel, Kadri Suija, Anna Tisler.

**Methodology:** Maali-Liina Remmel, Kadri Suija, Anda Kivite-Urtane, Mindaugas Stankūnas, Mari Nygård, Gunvor Aasbø, Laura Maļina, Anneli Uusküla.

**Project administration:** Maali-Liina Remmel.

**Supervision:** Kadri Suija, Anna Markina, Anneli Uusküla.

**Validation:** Anneli Uusküla.

**Visualization:** Maali-Liina Remmel.

**Writing – original draft:** Maali-Liina Remmel.

**Writing – review & editing:** Maali-Liina Remmel, Kadri Suija, Anna Markina, Anna Tisler, Anda Kivite-Urtane, Mindaugas Stankūnas, Mari Nygård, Gunvor Aasbø, Laura Maļina, Anneli Uusküla.

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
