## [Decision Letter · Decision Letter 0]

22 Nov 2024

PONE-D-24-42572Navigating the risks: stakeholder views on risk-based cervical cancer screening.PLOS ONE

Dear Dr. Remmel,

Thank you for submitting your manuscript to PLOS ONE. After careful consideration, we feel that it has merit but does not fully meet PLOS ONE’s publication criteria as it currently stands. Therefore, we invite you to submit a revised version of the manuscript that addresses the points raised during the review process.

We look forward to receiving your revised manuscript.

Kind regards,

Clement A. Adebamowo, BM, ChB Hons; FWACS, FACS, ScD, FASCO

Academic Editor

PLOS ONE

2. You indicated that ethical approval was not necessary for your study. We understand that the framework for ethical oversight requirements for studies of this type may differ depending on the setting and we would appreciate some further clarification regarding your research. Could you please provide further details on why your study is exempt from the need for approval and confirmation from your institutional review board or research ethics committee (e.g., in the form of a letter or email correspondence) that ethics review was not necessary for this study? Please include a copy of the correspondence as an "Other" file.

“This work was made possible by the funding received from the EEA (European Economic Area) and Norway Grants under grant EMP416.”

“This work was made possible by the funding received from the EEA (European Economic Ar-ea) and Norway Grants under grant EMP416.

The funding was obtained by Anneli Uusküla , Anda Kivite-Urtane ; Mindaugas Stankūnas ; Mari Nygård.

https://eeagrants.org/

Funders did not play a role in study design, data collection and analysis, decision to publish or preparation of the manuscript.”

Reviewers' comments:

Reviewer's Responses to Questions

**Comments to the Author**

1. Is the manuscript technically sound, and do the data support the conclusions?

Reviewer #1: Yes

Reviewer #2: Yes

2. Has the statistical analysis been performed appropriately and rigorously? 

Reviewer #1: N/A

Reviewer #2: N/A

3. Have the authors made all data underlying the findings in their manuscript fully available?

Reviewer #1: Yes

Reviewer #2: Yes

4. Is the manuscript presented in an intelligible fashion and written in standard English?

Reviewer #1: Yes

Reviewer #2: Yes

5. Review Comments to the Author

Reviewer #1: This novel study explores stakeholder views on personalised cervical screening in Estonia. This manuscript is very well-written and articulate, with a particularly strong discussion section. I have very minor comments but the clarification between feasibility and acceptability would serve to strengthen the aim of the paper.

Abstract:

Line 40: The abbreviation ‘RB’ has not been previously introduced, please include this on line 30 where the term risk-based is first used.

Introduction:

Lines 81-83: The focus of this study appears to be acceptability but the authors state that they are aiming to assess feasibility. Acceptability and feasibility are two distinct concepts, and I would avoid using these as synonyms. Based on the rest of the manuscript it seems that the authors are in fact aiming to assess acceptability and that, in the case of stakeholders, feasibility is a component of that.

Methods:

Lines 89-92: Please explain why you chose the TFA to develop your interview guide instead of the CFIR. I am curious as to why both of these frameworks have been used instead of just one, typically if a framework is used to inform the study design then the same framework is used in analysis.

Line 158: The data analysis section is clear and well-explained.

Line 174: Again, it would be beneficial to explain why CFIR was chosen at the analysis stage in favour of the TFA.

Results:

Lines 329-331: The comparison between different types of stakeholders, in this case midwives and gynaecologists, is very interesting. If such differences were observed elsewhere in the analysis, I would encourage drawing these out explicitly.

Discussion:

Overall, this is an extremely well-written discussion which contextualises your findings nicely.

Lines 360-363: This is an interesting and novel contribution to the literature.

Reviewer #2: Thank you for the opportunity to review this paper. I enjoyed reading it.

This paper takes a comprehensive approach to the analysis and reporting of its findings from 17 semi-structured interviews. The use of both frameworks, the Acceptability Framework to guide the development of the interview guide and CFIR for analysis, is nicely done and well-explained, and provides linked conclusions (plus the additional benefit of potential implementation strategies). The tables with quotes aligned to CFIR concepts provide clarity and transparency around findings. The results are clearly written and the discussion insightful and relevant to the international interest in risk-based screening. Recommended for publication with minor revision.

Comments:

1. Line 68: Please add some references to examples (ref 32,33 etc) of existing research

2. The term ‘stakeholder’. The authors rightly acknowledge that community/patients may not be seen as a priority stakeholder group in the study’s context. However, by definition and in other contexts they are. I think ‘Stakeholder’ is acceptable for the Title but suggest clarifying ‘which stakeholders’ earlier in the paper and within Figure 1 to avoid confusion, e.g.,

Line 82: add ‘healthcare service and policy level stakeholders’

Figure 1: Suggest make title clearer or highlight that concepts related to “Individuals-women” are from the perspective of healthcare policy and service level providers, so this figure can stand alone.

3. Line 194-195: Check format of SD is correct i.e., suggest (SD=8.6)

4. Line 266: What is the relevance of digital literacy in this context? Was there a link to inequalities related to digital literacy in your tables? Please add a sentence for clarity. Consider also in the Discussion (Line 402).

5. Line 282: Headings in Table 3 (‘Inner setting and Outer setting’) are back-to-front. Please review and correct.

6. Figure 1: This looks attractive and potentially useful for researchers and planners. Suggest but not required to combine’ Innovation’ heading/graphic and concepts as one box, and ‘Implementation’ heading/graphic and concepts similarly to reduce complexity. See comments above re: title and women

6. PLOS authors have the option to publish the peer review history of their article (what does this mean? ). If published, this will include your full peer review and any attached files.

**Do you want your identity to be public for this peer review?** For information about this choice, including consent withdrawal, please see our Privacy Policy .

Reviewer #1: No

Reviewer #2: No

---

## [Author Response · Author response to Decision Letter 1]

31 Dec 2024

Dear Editor Dr. Clement A. Adebamowo,

I would like to thank you for the opportunity to submit a revised manuscript titled “Navigating the risks: stakeholder views on risk-based cervical cancer screening.” to PLOS ONE.

We appreciate the time and effort that you and the reviewers have dedicated to providing your thoughtful and constructive feedback on our manuscript.

We believe that we have addressed all the concerns raised in the review and that the manuscript has been strengthened as a result.

Our responses to the review comments are presented below. All changes made in the manuscript are highlighted as track changes.

Thank you for your consideration of this manuscript.

We look forward to hearing from you regarding our submission and are ready to address any further questions and comments you may have.

Best regards,

Maali-Liina Remmel

Editorial Comments

1. Request

Answer

The manuscript has been reviewed and corrected as per PLOS ONE's style requirements, including those related to file naming.

2. Request

Could you please provide further details on why your study is exempt from the need for approval and confirmation from your institutional review board or research ethics committee (e.g., in the form of a letter or email correspondence) that ethics review was not necessary for this study? Please include a copy of the correspondence as an "Other" file.

Answer

We appreciate the opportunity to address the editor's request for further details regarding the ethical considerations of our study. Ethical approval was not sought as the study qualifies for exemption according to the applicable ethics committee regulations and Estonian legislation.¹,²

Specifically, the study:

(i) did not collect medical data;

(ii) did not involve patients;

(iii) focused on professional knowledge within participants’ public roles;

(iv) did not involve sensitive data; and

(v) posed minimal risk to participants.

The study complied with the ethical guidelines set forth in the Belmont Report. All data were de-identified. Informed consent was obtained after participants were fully briefed on the study's purpose, procedures, their right to withdraw, potential risks or discomforts, possible benefits, confidentiality safeguards, and contact details.

While the University of Tartu’s ethics committee does not provide formal confirmations of exemption³, similar studies typically do not require ethical approval in other jurisdictions (e.g., the UK’s NHS Health Research Authority and NIH Grant/funding decision tools⁴; results attached as an Other file).

Several categories of research are typically exempt from ethics review. A recent review of national guidance in developed countries, including Australia, the United Kingdom, the United States, and the Netherlands, identified nine such categories. One common exemption is “research with staff in their professional role,” which applies to our study.⁵

We believe this clarifies the ethical considerations of our study and its compliance with relevant regulations.

References for Comment 2

¹ Estonian Personal Data Protection Act. [Accessed 12.12.24]

² University of Tartu Guide for data protection in Research. [Accessed 12.12.24]

³ Research Ethics Committee of the University of Tartu. [Accessed 12.12.24]

⁴ HRA decision tool. [Accessed 09.12.24]

⁵ Scott AM, Kolstoe S, Ploem MCC, Hammatt Z, Glasziou P. Health Res Policy Syst. 2020;18(1):11. doi: 10.1186/s12961-019-0520-4.

3. Request

Please remove any funding-related text from the manuscript and let us know how you would like to update your Funding Statement.

Answer

This text has been removed and no updates are required for the current Funding Statement.

4. Request

We note that you have indicated that there are restrictions to data sharing for this study. For studies involving human research participant data or other sensitive data, we encourage authors to share de-identified or anonymized data. However, when data cannot be publicly shared for ethical reasons, we allow authors to make their data sets available upon request. Please address the following prompts:

a) If there are ethical or legal restrictions on sharing a de-identified data set, please explain them…

b) If there are no restrictions, please upload the minimal anonymized data set…

Answer

We acknowledge the importance of addressing this matter.

a) Even if deanonymized, indirect identifiers may risk the identification of individuals participating in this study. Therefore, to protect their confidentiality, the data will not be publicly shared. However, deanonymized data could be shared upon reasonable request. The data are in the Estonian language. Requests for information regarding data should be sent to Maali-Liina Remmel at maali-liina.remmel@ut.ee.

5. Request

Answer

We have carefully reviewed the reference list to ensure it is complete and accurate.

Reviewer #1

Abstract

Request

Line 40: The abbreviation ‘RB’ has not been previously introduced, please include this on line 30 where the term risk-based is first used.

Answer

The abbreviation ‘RB’ has now been introduced where the term “risk-based” is first mentioned. We appreciate your attention to detail.

Introduction

Request

Lines 81-83: The focus of this study appears to be acceptability but the authors state that they are aiming to assess feasibility. Acceptability and feasibility are two distinct concepts, and I would avoid using these as synonyms. Based on the rest of the manuscript it seems that the authors are in fact aiming to assess acceptability and that, in the case of stakeholders, feasibility is a component of that.

Answer

We appreciate the reviewer’s insightful comment and fully agree that acceptability and feasibility are distinct concepts. Upon reflection, we recognize that the primary focus of our study is indeed on assessing the acceptability of risk-based cervical cancer screening (RB CCS) programs, with feasibility being a component within that broader concept. We have revised the manuscript accordingly.

Methods (Lines 89-92)

Request

Please explain why you chose the TFA to develop your interview guide instead of the CFIR. I am curious as to why both of these frameworks have been used instead of just one, typically if a framework is used to inform the study design then the same framework is used in analysis.

Answer

Thank you for this insightful question. While a single framework often guides both design and analysis, our study’s needs led us to utilize both TFA and CFIR. TFA allowed an in-depth focus on acceptability—our primary interest—by directly addressing factors like affective attitude, burden, ethicality, etc. However, we also recognized the importance of the broader implementation context, so we drew on CFIR to capture factors beyond individual acceptance.

During interviews, participants frequently discussed issues tied to healthcare infrastructure, policy context, and implementation processes, which fit better under CFIR’s constructs (inner setting, outer setting, process). This dual-framework approach gave us a comprehensive view, balancing individual-level acceptance with the broader environment influencing risk-based cancer screening. We have clarified this rationale in the manuscript.

Methods (Line 158)

Request

The data analysis section is clear and well-explained.

Answer

Thank you for your positive feedback on the data analysis section.

Methods (Line 174)

Request

Again, it would be beneficial to explain why CFIR was chosen at the analysis stage in favour of the TFA.

Answer

Clarifications were added to the manuscript.

Results (Lines 329-331)

Request

The comparison between different types of stakeholders, in this case midwives and gynaecologists, is very interesting. If such differences were observed elsewhere in the analysis, I would encourage drawing these out explicitly.

Answer

Thank you for highlighting this point. Beyond the differences explicitly noted in the manuscript, no other clearly distinguishable differences were observed among stakeholder groups during the analysis.

Discussion

Request

Overall, this is an extremely well-written discussion which contextualises your findings nicely.

Lines 360-363: This is an interesting and novel contribution to the literature.

Answer

Thank you very much for your kind feedback on our discussion section.

Reviewer #2

1. Request

Line 68: Please add some references to examples (ref 32,33 etc.) of existing research.

Answer

Thank you for your suggestion. We have now added relevant references to illustrate existing research and to support our statement.

2. Request

The term ‘stakeholder.’ The authors rightly acknowledge that community/patients may not be seen as a priority stakeholder group in the study’s context. However, by definition and in other contexts they are. I think ‘Stakeholder’ is acceptable for the Title but suggest clarifying ‘which stakeholders’ earlier in the paper and within Figure 1 to avoid confusion, e.g.,

• Line 82: add ‘healthcare service and policy level stakeholders’

• Figure 1: highlight that concepts related to “Individuals-women” are from the perspective of healthcare policy and service level providers, so this figure can stand alone.

Answer

Thank you for this thoughtful feedback. We agree and have clarified “healthcare service and policy-level stakeholders” on line 82. We also revised the title and included a note within Figure 1 to indicate that references to “Individuals-women” represent perspectives from healthcare policy and service-level providers.

3. Request

Line 194-195: Check format of SD is correct i.e., suggest (SD=8.6).

Answer

Thank you for pointing this out. We have revised the format accordingly.

4. Request

Line 266: What is the relevance of digital literacy in this context? Was there a link to inequalities related to digital literacy in your tables? Please add a sentence for clarity. Consider also in the Discussion (Line 402).

Answer

Thank you for your comment. Digital literacy emerged as a potential barrier to equitable participation in risk-based screening programs. Stakeholders raised concerns that some participants may struggle with digital tools, such as online risk assessments, leading to inequalities. We have added a clarifying sentence in the Results and Discussion sections.

5. Request

Line 282: Headings in Table 3 (‘Inner setting and Outer setting’) are back-to-front. Please review and correct.

Answer

Thank you for pointing this out. We have corrected the headings in Table 3 to ensure proper alignment with their respective content.

6. Request

Figure 1: This looks attractive and potentially useful for researchers and planners. Suggest, but not required, to combine ‘Innovation’ heading/graphic and concepts as one box, and ‘Implementation’ heading/graphic and concepts similarly to reduce complexity. See comments above re: title and women.

Answer

Thank you for your positive feedback. We have merged the “Innovation” and “Implementation” concepts into single boxes to reduce complexity and have addressed the title/women clarification as suggested.

---

## [Editor Report · Decision Letter 1]

9 Jan 2025

Navigating the risks: stakeholder views on risk-based cervical cancer screening.

PONE-D-24-42572R1

Dear Dr. Remmel,

We’re pleased to inform you that your manuscript has been judged scientifically suitable for publication and will be formally accepted for publication once it meets all outstanding technical requirements.

Kind regards,

Clement A. Adebamowo, BM, ChB Hons; FWACS, FACS, ScD, FASCO

Academic Editor

PLOS ONE
---

## [Editor Report · Acceptance letter]

PONE-D-24-42572R1

PLOS ONE

Dear Dr. Remmel,

I'm pleased to inform you that your manuscript has been deemed suitable for publication in PLOS ONE. Congratulations! Your manuscript is now being handed over to our production team.

Kind regards,

on behalf of

MR Clement A. Adebamowo

Academic Editor

PLOS ONE